# Smart and low-cost fluorometer for identifying breast cancer malignancy based on lipid droplets accumulation

Shiva Moghtaderi[1]*, Aditya Mandapati[2], Gerald Davies[2], Khan A. Wahid[1], Kiven Erique Lukong[2]

1 Department of Electrical and Computer Engineering, University of Saskatchewan, Saskatoon, Saskatchewan, Canada, 2 Department of Biochemistry, Microbiology and Immunology, University of Saskatchewan, Saskatoon, Saskatchewan, Canada

* shiva.moghtaderi@usask.ca

**Data Availability Statement:** All data are fully available without restriction. All relevant data are

## Abstract

The most common cause of breast cancer-related death is tumor recurrence. To develop more effective treatments, the identification of cancer cell specific malignancy indicators is therefore critical. Lipid droplets are known as an emerging hallmark in aggressive breast tumors. A common technique that can be used for observing molecules in cancer microenvironment is fluorescence microscopy. We describe the design, development and applicability of a smart fluorometer to detect lipid droplet accumulation based on the emitted fluorescence signals from highly malignant (MDA-MB-231) and mildly malignant (MCF7) breast cancer cell lines, that are stained with BODIPY dye. This device uses a visible-range light source as an excitation source and a spectral sensor as the detector. A commercial imaging system was used to examine the fluorescent cancer cell lines before being validated in a preclinical setting with the developed prototype. The outcomes indicate that this low-cost fluorometer can effectively detect the alterations levels of lipid droplets and hence distinguish between "moderately malignant" and "highly malignant" cancer cells. In comparison to prior research that used fluorescence spectroscopy techniques to detect cancer biomarkers, this study revealed enhanced capability in classifying mildly and highly malignant cancer cell lines.

## 1. Introduction

The treatment approaches for breast cancer are guided by molecular features [1]. The first line of treatment for hormone receptor-positive (HR+) cancer patients is hormone treatment with the estrogen blocker tamoxifen, treated with aromatase inhibitors such as letrozole and/or fulvestrant, a selective estrogen receptor degrader, as second-line treatments. The standard treatment for human epidermal receptor (HER2+) breast cancer patients is a combination therapy of anti-HER2 monoclonal antibody and chemotherapy. The triple negative breast cancer (TNBC) lacks targeted treatment options and is therefore most challenging compared to HER2+ and HR+ cases. The standard treatment for TNBC is chemotherapy [2]. The location, stage and type of cancer typically determine the next course of treatment [3]. Despite these

within the paper and its Supporting information files.

**Funding:** The New Frontiers in Research Fund (NFRF). The funders had no role in study design, data collection and analysis, decision to publish, or preparation of the manuscript.

advances in the treatment of primary breast tumors, all breast cancer subtypes are prone to metastases. Therefore, encountering metastasis and tumor recurrence at local or distant regions is common among breast cancer survivors [4]. Understanding the molecular and cellular mechanisms underlying metastasis and identifying reliable biomarkers for malignant breast cancer remain an active research area in recent years.

Metabolic reprogramming in cancer cells is regarded as one of the major hallmarks of the cancer [5]. A well-characterized part of this process involves abnormal glucose metabolism through the Warburg Effect, enabling cancer cells to spread rapidly through aerobic glycolysis [6]. However, latest evidence is progressively demonstrating the significance of lipid metabolism in the development and progression of tumors [7]. Lipid droplets, the multifunctional organelles, are associated with metabolic activities, signaling, and the creation of inflammatory mediators [8, 9]. For cancer cell proliferation, resistance to death, and malignancy, lipid droplet formation and catabolism are essential. They are also closely linked to energy metabolism and cell signaling [8, 9]. The lipid droplets' metabolic issues are connected to numerous metabolic related illnesses for example fatty liver, diabetes, obesity, and cardiovascular disorders [8, 10–12]. The deposition of lipid droplets in non-adipocyte tissues has been proposed as a novel cancer signature [8]. Increased lipid droplet concentrations have been observed in cancer cells and diseased organs when compared to noncancerous cells. Reported cancers including colorectal cancer, prostate and breast cancers, renal cell carcinoma, hepatocellular carcinoma and glioblastoma [8]. Lipids are crucial parts of the cellular system that controls the migration, proliferation and protecting cancerous cells. That being said lipids perform a variety of roles in membranes morphology, cellular signaling, and protein regulation [13]. Specifically, lipid droplets have been observed to initiate or control these behaviors in cancer cells [14]. Researchers discovered that the number of lipid droplets in a panel of breast cell lines corresponds with aggressiveness [8]. According to a number of research, malignant breast cancer cells have more intracellular lipid droplets [8]. Also, it has been illustrated that lipid droplets contain proteins engaged in breast cancer metastasis and invasion, which are responsible for promoting malignancy and progression [15].

In lipid droplets, there is only one layer of phospholipids encasing a hydrophobic core of neutral lipids and proteins which is based on a cell's demands [9]. Indeed, in cells undergoing rapid rates of proliferation, there is a noticeable rise in the production of lipid droplets [9]. In fact, cell cycle arrest is caused by the reduction of lipid droplets in renal cancer cells [16]. Interestingly, a comparison of lipid droplet content in highly malignant (MDA-MB-231) and mildly malignant (MCF7) breast cancer cells has shown that the more malignant and aggressive MDA-MB-231 cells contain a higher density of lipid droplets [13]. The results of this study show that the density of lipid droplets (LDs) in normal breast epithelial cell line MCF10A is 50% lower than LD density in low-malignancy breast cancer cells (MCF7) [13]. Highly malignant MDA-MB-231 cells have approximately 4 times as much LD density as MCF10A cells [13]. Therefore, a higher density of lipid droplets is linked a more aggressive kind of cancer [13].

Lipid droplets play a significant role in mediating the proliferation, invasion, metastasis, and resistance to chemotherapy in a number of cancer types as a sophisticated, dynamic, multifunctional, and storage organelle [8]. Numerous studies have been conducted to discover variations in the lipid droplet biochemistry and link them to various stages of breast cancer [13]. In patients with advanced cancer, the formation of lipid droplets may be used as a potential indicator of how they would respond to immunotherapies or traditional neoadjuvant therapy [8]. With the emerging role of lipid droplets in breast cancer and the malignancy of aggressive breast cancer cases, identifying a method of detection for lipid droplets may be an effective way to identify fast growing tumor cells.

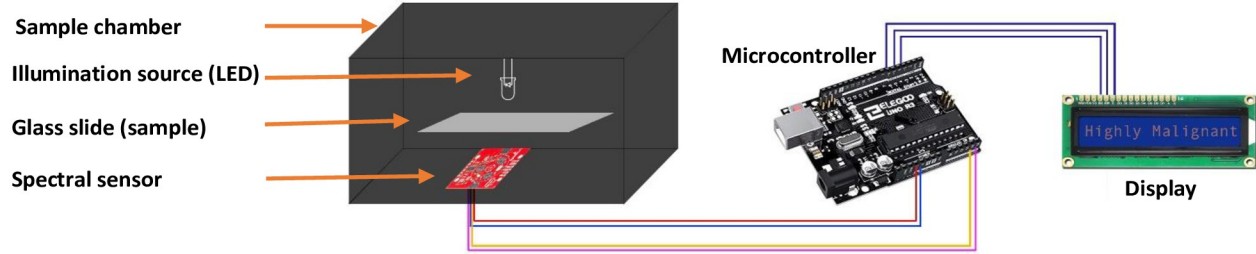

**Fig 1. Graphical illustration of the proposed instrument.**

Fluorescence imaging is a non-invasive and non-toxic imaging method used to visualize biological molecules and processes using fluorescent dyes or proteins as markers. Fluorescence imaging is in fact fast and frequently utilised for identification of biological samples procedures [17, 18]. Modern labs frequently use fluorescence microscopy to examine, locate, and monitor individual fluorescing particles [19]. Imaging at the molecule level allows for disease-causing physiological pathways investigation rather than conventional medical imaging methods such as MRI, CT and ultrasound [20]. By using cancer-specific fluorescence probes, the tumor shows a fluorescence signal much higher than the background autofluorescence signal emitted by healthy tissues. Researchers can extract useful information from fluorescently labelled proteins of interest. Numerous fluorescence imaging devices for biological applications are commercially available by various firms across the world [21]. Multiple processes are required for microscopic analysis of tissue samples which are time-consuming tissue slide preparation, comprising formalin fixation, paraffin embedding, slicing and staining [22]. However, these devices are usually expensive and bulky and require multiple sophisticated equipment and lab environment. Furthermore, the measurements from these devices may also be susceptible to unreliable variation caused owing to the changing environment/atmosphere, detectors, instrument layout as well as the calibration of the hardware [23]. Fluorometers are also frequently used in laboratories to detect fluorescent signals emitted by the subject of interest. The emitted signal is used to identify the presence and the number of specific molecules in a medium. However, the cost of the ultrasensitive imaging equipment necessary for detecting the fluorescence signal is high and the entire imaging process is reliant on sophisticated hardware and software [23].

In this paper, a low-cost and smart fluorometer is designed and developed which is able to distinguish different levels of lipid droplets in fluorescently stained MDA-MB-231 and MCF7 cell lines. Designing a fluorescence-based monitoring device that can detect lipid droplet levels and discriminate between moderately and highly malignant breast cancer cells has not previously been the subject of study. Our proposed fluorescence detection instrument is shown in Fig 1. On both sides of the sample holder, the excitation source and the sensor are positioned in a straight line to detect lipid droplets levels in breast cancer cells lines.

## 2. Materials and methods

The major purpose of this study is to show that our instrument can detect varying quantities of light emitted by highly malignant and mildly malignant cell lines as a result of variable levels of lipid droplets. However, it is difficult to measure lipid droplet accumulation differences in a reliable manner because the circumstances which cells are grown may have an impact. To assess the presence and the amount of the lipid-rich regions by fluorometer, measurements

were performed on the cell samples that were fluorescently-labelled with boron-dipyrro-methene (BODIPY) dye.

To identify lipid droplets, BODIPY, a fluorescent dye that binds specifically to lipid droplets, can be used [24, 25]. The goal of applying BODIPY dye is to target lipid droplets (LDs) specifically in these cells and, through quantification of fluorescence intensity, measure the accumulation of LDs as a measure of malignancy [26, 27]. BODIPY is inherently lipophilic and fluoresecent. In our experiment, the aim was to excite the BODIPY dye with the proper wavelength of the light. Consequently, detect accumulation of lipid droplets in highly malignant cancer cell line (MDA-MB-231) and the mildly malignant control cell line (MCF7), with non-cancerous mammary cells MCF10A as a control.

### 2.1 Analyzing wavelengths

To build an efficient fluorescence optical system for medical purposes, a comprehensive understanding of fluorescence is required. Fluorescent labelling is the covalent attachment of fluorescent dyes to biomolecules like proteins. This makes them distinctive from non-fluorescently labelled molecules [3, 28]. Fluorophores absorb light and re-emit it at longer wavelengths. This process provides an opportunity to visually detect various activities in living cells and tissues. Fig 2 shows the absorbance and emission spectra of the BODIPY dye.

When choosing the excitation source, the target fluorophore's absorption/emission spectra were considered. The breast cancer cells were conjugated with BODIPY that has a peak excitation and emission wavelength of 500 nm and 520 nm respectively.

### 2.2 Sample preparation

BODIPY-stained MCF7, MCF10A and MDA-MB-231 slides for fluorescent microscopy and detection using the sensor were prepared as stated in [26]. Briefly, autoclaved coverslips were treated with poly-L-lysine to promote cell adherence before being seeded with cells. When the

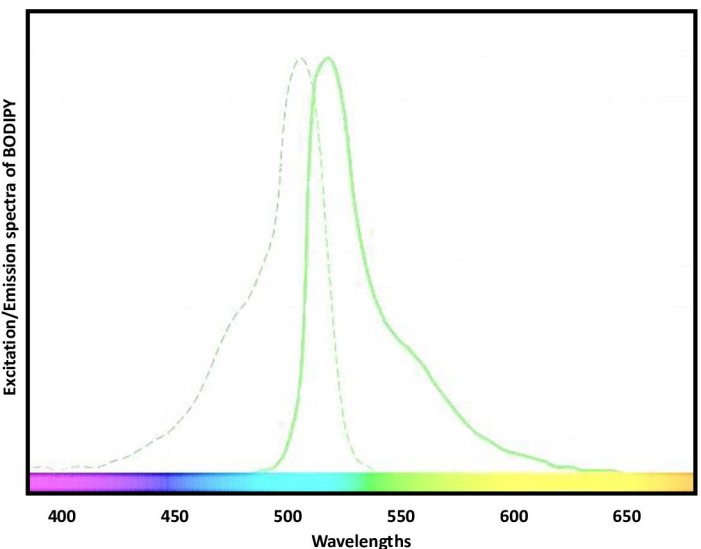

**Fig 2. BODIPY's excitation and emission spectra.** The peak absorption of 500nm and peak emission of 520nm for the cultured sample are shown in this spectra [29].

cells reached a confluency of 50%, they were stained with a 2 μM solution of BODIPY in Phosphate-Buffered Saline (PBS) for 15 mins at 37 degrees Celsius.

PBS was used to wash the coverslips three times. Then it is fixed in 4% paraformaldehyde at room temperature for 30 mins. After three PBS washes, they were mounted using ProLong Gold antifade mounting solution on glass slides(with DAPI) (ThermoScientific, P36931). Finally, the slides were allowed to cure overnight, at 4 degrees Celsius in the dark.

### 2.3 Design of the lab prototype

Acquiring excellent sensitivity while also achieving low power consumption is the toughest challenge for building a portable and affordable fluorometer [30]. In other words, the lower price point is usually at the expense of greater specialization, less efficiency or worst sensitivity and throughput [30]. The intensity of the emitted fluorescence signal and the sensitivity of the detector utilized in the measurement setup, are two critical factors for designing a successful fluorescence-based cancer diagnosis instrumentation. That is why component selection is essential in designing a fluorescence detection system.

In this paper, the performance of the device with sample positioned between the LED and the spectral sensor is investigated to verify the system's ability to detect emitted fluorescence signal.

Fig 1 shows the working principle of the proposed fluorometer. As visible, the top of the sample is the focal point of the light emitting diodes (LED) excitation light. We mounted the excitation source extremely close to the sample because the LED light was doughnut-shaped and we wanted to make sure that the light was focused on the sample. The LED emits light with a peak wavelength of 500 nm in the visible spectrum. The detector receives the longer wavelength light which the sample emits.

The AS7265 spectral sensor, made by AMS, is used to detect fluorescent light. AS7265 delivers 18-channel multi-spectral sensing in the visible wavelengths between 410nm and 940nm with a full-width half-max (FWHM) of 20nm. We selected AS7265 as the detector because Gaussian filters are integrated into each channel. It enables the control of the light entering the sensor array using an integrated filter. As a result, there is no need for a filter to be placed between the sample and the detector. This will reduce the cost and complexity of the fluorometer.

The device is integrated in a black box that creates a dark environment preventing outside light from interfering. The circuit is completely contained in the box, allowing the entire system to be powered by a micro-USB cable and a standard 5-volt battery bank. After the device is initialized, the sensor will receive the emitted signal from the sample. The microcontroller transmits the emission signal to the computer through the I2C port. Based on the dye's features, we expect that fluorescence emission signal to be at 520nm. As such, the E and F channels are selected for the detection.

Fig 3 depicts the breast cancer cell lines. It can be observed that these cell lines are not distinguishable with the unaided eye. However, when we monitor the same cell lines with the proposed device (Fig 4), we can distinguish between these cells. It is clear from Fig 5 that the reflected fluorescent response from MDA-MB-321 substantially exceeds the emission from the MCF7. The result proves that the designed instrument is able to distinguish between different amounts of emitted signals related to the accumulation of lipid droplets.

## 3. Results and discussion

During the experiment, we assess the capability of the proposed device to detect different levels of fluorescence emission signals caused by the varying amounts of lipid droplets. At this point,

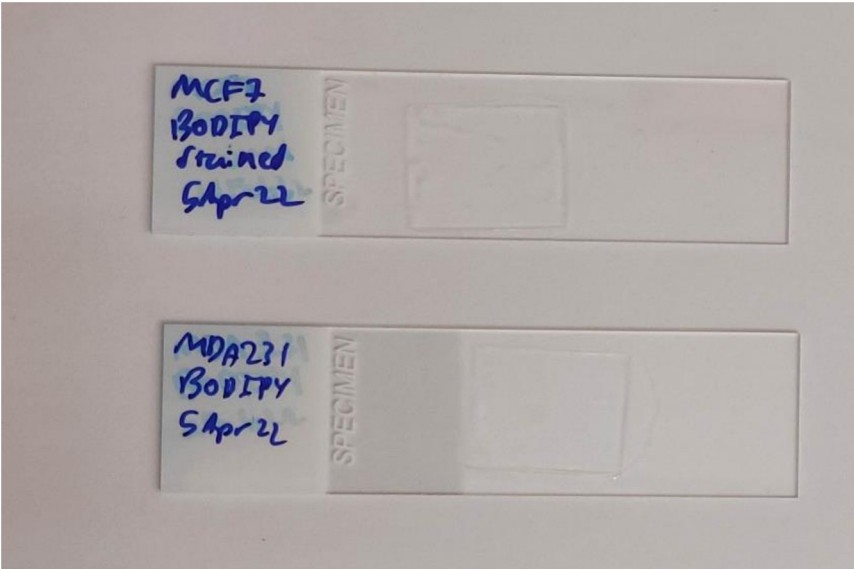

**Fig 3. Slides MCF7 and MDA-MB-231 breast cancer cells were fixed on coverslips and treated with the fluorescent dye BODIPY.**

the purpose was to ensure that the selected wavelength for the illumination source, could properly excite the dye. In other words, we have studied the emitted fluorescence signal under different LED wavelengths to compare the resulting signals from twenty MDA-MB-231 slides and twenty MCF7 slides, with twelve MCF10A slides as non-cancerous controls. The average fluorescence intensity of 52 slides captured by the spectral detector under varying excitation wavelengths is observed in Fig 5.

We found that when the excitation wavelengths are 465nm, 475nm and 525nm, both samples (MDA-MB-231 and MCF7) emit relatively same signals. But when these samples are exposed to 500nm light, the dye is excited and there is a distinguishable difference when compared by value. This may be the main difference between MDA-MB-231 and MCF7 that is detected by the proposed device. Both cell lines also fluoresced significantly greater than MCF10A under 500nm excitation, as was expected.

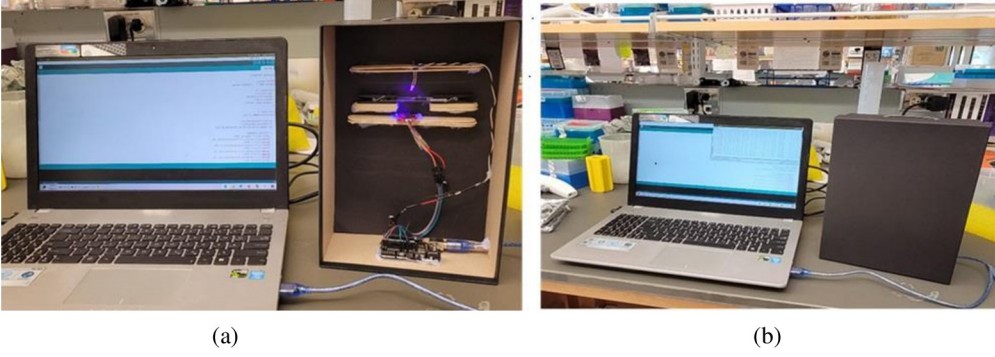

|                     (a)                     |                     (b)                     |

**Fig 4.** Experimental setup: (a) proposed instrument. (b) The device is integrated in a black box that creates a dark environment preventing outside light.

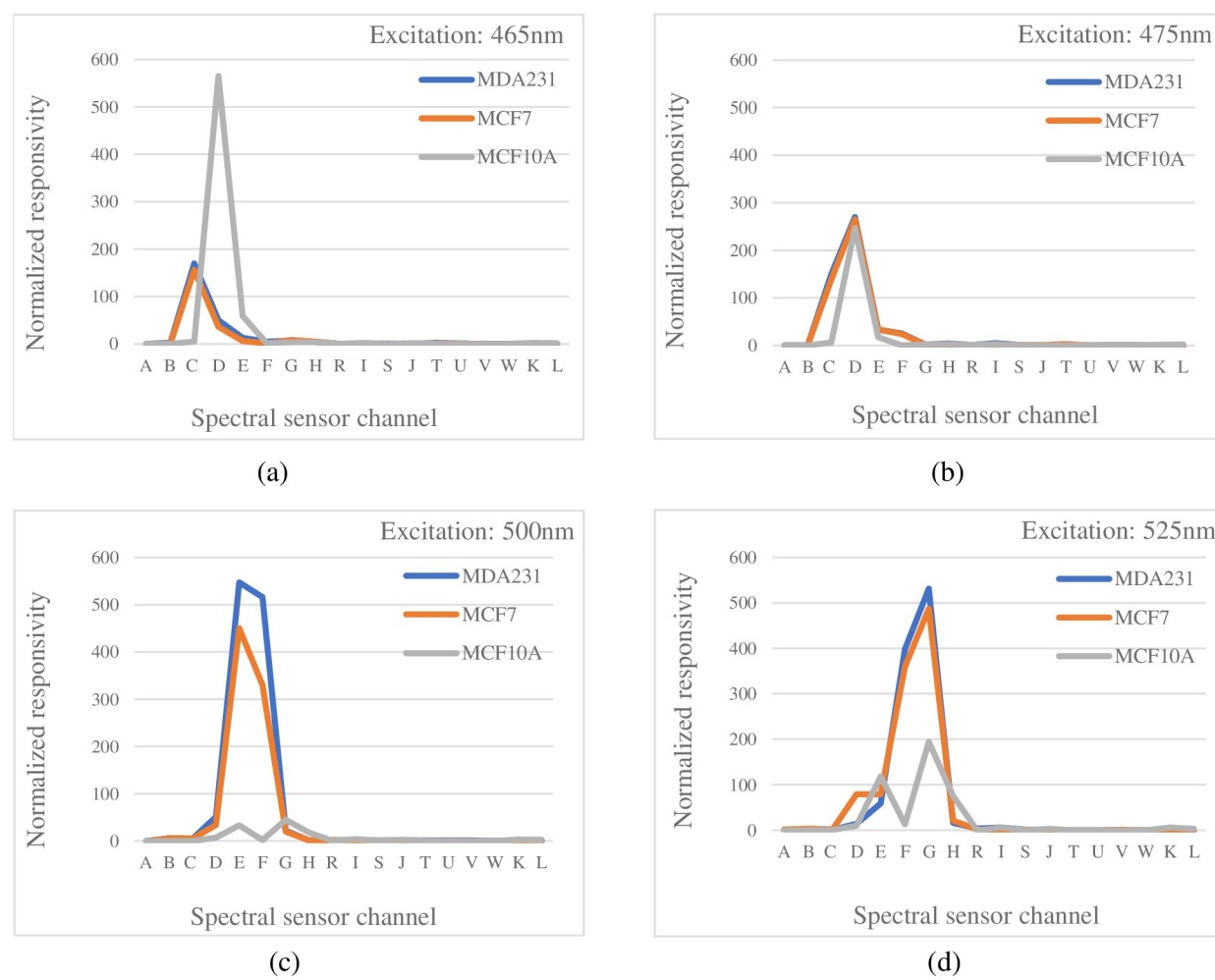

**Fig 5.** Emission signal resulted from BODIPY stained MCF7, MDA-MB-231 and MCF10A when excited under (a) 465nm, (b) 475nm, (c) 500nm and (d) 525nm LED.

In the second experiment, fluorescence microscopy was deployed to investigate distinctive amounts of lipid droplets. Fig 6 shows the cultured MDA-MB-231 and MCF-7 cell lines expressing BODIPY viewed under a commercial imaging system (ThermoFisher Scientific EVOS M5000 Imaging System). The images clearly exhibit a distinctive fluorescence signature and remarkably different emission signal responsivity between highly malignant and mildly malignant breast cancer cell lines. The same cell lines were utilised in the first and second experiments with the proposed instrument and the commercial system to validate the reading.

The third experiment was performed on totally unstained MDA-MB-231 and MCF-7 cell lines to check for any inherent autofluorescence originating from the sample itself or the sample matrix. When biological substances are activated by a particular wavelength, they emit light in the UV-visible or near-IR spectral range. This phenomenon is recognized as native fluorescence or autofluorescence[31].

Autofluorescence can be caused by a variety of factors, and numerous sources might exist within a single sample. Endogenous autofluorescence can be even produced by cell preparations steps. The way the samples are processed before testing can also introduce additional

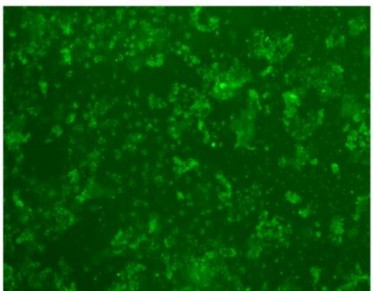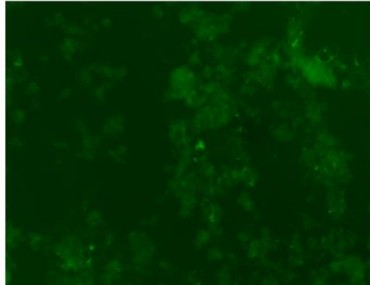

**Fig 6.** Cultured (a) highly malignant (MDA-MB-231) and (b) mildly malignant (MCF7) cell lines has expressing BODIPY viewed under a commercial imaging system.

background fluorescence. As a result, while we are focusing on targeted fluorescence, we also need to address autofluorescence. It's critical to know the spectra of the possible autofluorescence signal. Fig 8 demonstrates emission signal resulted from unstained MCF7 and MDA-MB- 231.

The proposed device was used to extract the autofluorescence spectrum. Spectral scanning will improve experiment optimization and accuracy. This will ensure us that any autofluorescence related strong peaks are avoided in the main experiment.

In this work, we performed a test on unstained MDA-MB-231 and MCF-7 cell line with the proposed device. The resultant signal is presented in Fig 7. As observed, no major signal is detected by the spectral sensor, while the unstained samples are illuminated with a 500nm LED. Additionally, both cancerous samples (MCF7 and MDA-MB-231) produced a grater signal than the non-cancerous control MCF10A.

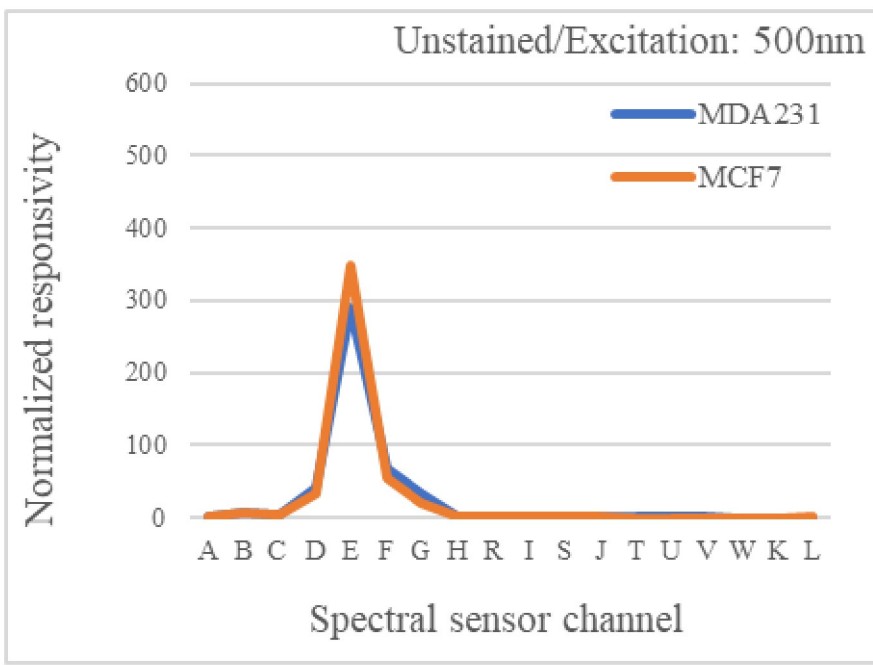

**Fig 7. Emission signal resulted from unstained MCF7 and MDA-MB- 231 when excited under 500nm LED.**

In our last experiment, we employed measuring approaches including sensitivity, specificity, accuracy and precision that are typically used in classification to assess the performance of the proposed instrument. Ten samples (four of MDA-MB-231 expressing BODIPY and six of MCF7 expressing BODIPY) were cultured on the glass slide. The slides were evaluated with the ThermoFisher imaging system and then the proposed instrument was applied.

The provided device was capable of accurately detecting all the highly malignant cell lines from the highly malignant samples. The fluorimeter spotted 3 out of 6 mildly malignant cell lines correctly. The findings revealed that the sensitivity, accuracy, specificity, and precision are 50%, 70%, 100% and 100% respectively. Fig 8 demonstrates the confusion matrix drawn from the measurements obtained with our prototype.

The production and accumulation of lipid droplets is a direct function of a cancer cells metabolism [9]. There may be fluctuations in the quantity of lipids produced by a cell depending on growth conditions as well as the stage of the cell cycle any given cell is in, at the point of observation [9]. Therefore, measurement of reliable differences in lipid droplet accumulation is challenging because it could be impacted by variables in which cells are grown as well as the timing of observation. Additionally, the limitations of BODIPY as a dye must be noted. Mainly, the use of the dye is limited by its photostability and background noise, though it can reliably stain LDs and can permeate cells effectively [32].

In this work, we put our fluorometer to the test by analyzing the emitted fluorescence light from breast cancer cell lines that had been genetically modified to express the BODIPY. For the fluorescence-based selective detection of breast cancer cells, these cells acted as a biologically acceptable and technically convenient representative of patient tissue.

Our fluorometer prototype was created to detect fluorescence emitted by cancer cell lines in vitro. The fluorometer output was unaffected by the morphology of the cancer cells or the dimensions of the glass slides and coverslips. These parameters were all uniform during the experiment.

The resulting device can successfully differentiate between high and low lipid droplets concentration in MDA-MB-231 and MCF-7 cell lines while successfully differentiating both from the non-cancerous MCF10A cells. This device is simplified from a conventional fluorescence

| | | Predicted condition | |
|---|---|---|---|
| | | Highly malignant | Mildly malignant |
| Real condition | Highly malignant | 4 <br> True negative (TN) | 0 <br> False positive (FP) |
| | Mildly malignant | 3 <br> False negative (FN) | 3 <br> True positive (TP) |

**Fig 8. Confusion matrix.**

microscope as it has less complexity. The results from the spectral sensor will be sent to the computer through the I2C connection in real-time. The connection of the fluorometer uses extremely little memory and operates well on computer [19].

The potential applications of our proposed fluorometer extend prominently to the domains of early detection and screening. This include complementing the advancements in wireless capsule endoscopy that introduces a new dimension of assessing tissue characteristics by considering the lipid droplets accumulation. In the context of early detection and screening, our fluorometer's ability to detect cancer malignancy through lipid droplet accumulations holds promise for providing a non-invasive and cost-effective method for identifying cancerous cells at an early stage. This can play a crucial role in identifying potential cases for further accurate evaluations.

It is also important to understand that the application of the proposed device should be assessed in light of both the intended use and the larger clinical landscape. Our device's affordability and portability make it suitable for deployment in regions with limited resources. In such settings, even a device with slightly lower sensitivity can play a crucial role in identifying potential cases for further accurate evaluations, thus contributing to early intervention. Moreover, in cases where frequent monitoring is required, such as post-treatment follow-up, a less sensitive device may still provide valuable insights. It can help in tracking changes over time and alerting healthcare providers to any deviations from the baseline.

As previously mentioned, in cancer research, lipid metabolism is a novel concept. Growing data suggests that lipid biomarkers are an important cancer indication [13, 33–36]. Recent results are indicating an even tighter connection between lipid metabolism and malignancy through different technical and methodological procedures.

We have compared the efficiency and features of our device with comparable designs in Table 1. As observed, the progress in fluorescence microscopy and molecular tagging in different types of cancer cell lines, has been verified. Compared to other similar studies using fluorescence spectroscopy techniques for detecting cancer biomarkers, our study demonstrated enhanced ability in classification of mildly malignant and highly malignant cancer cell lines. In our study model, we conducted a test to demonstrate that it is possible to detect different levels of lipid droplets using a low-cost and compact fluorescence-based detection instrument. This

**Table 1. Comparison with other devices.**

| Work | Device | Goal | Cancer/Specific target | Dye | Sensitivity | Cost $USD |
|------|--------|------|------------------------|-----|-------------|-----------|
| [23] | Fluorometer | Capable of discriminating between cancer and control cells. | Breast cancer cells | GFP | 100% | $138.38 |
| [37] | Fluorometer | Can distinguish between the cancer cells and control cells. | Colorectal Cancer Cells | IRFP702 | 85% | $224.37 |
| [38] | Fluorescence microscopy | System has potential application in cancer gene diagnosis. | Tumor suppressor/ p53 cancer gene | Fluorescein | - | - |
| [39] | Fluorescence microscopy | Detect small tumor lesions or tumor-bearing LN. | Pancreatic cancer/ Epidermal growth factor receptor | IRDye800 | - | - |
| [30] | Fluorescence microscopy | Monitors biochemical reactions on disposable paper strips | Cas13a RNA | 1x SybR Green II | - | $15 |
| [40] | Fluorometer | Cancer detection and bladder cancer grading. | Bladder cancer/ volatile organic compounds (VOCs) | Porphyrins, metalloporphyrins, pH indicators, solvatochromic | 84.21% for cancer detection /66.67% for grading | $5.26 |
| **This work** | Fluorometer | Can effectively detect the lipid droplets levels' alterations and distinguish between moderately and highly malignant cancer cells. | Breast cancer cells/ Lipid droplets | BODIPY | 50% | $104.46 |

**Table 2. The proposed fluorometer's costs.**

| Component | Model | Estimated cost ($USD) |
|---|---|---|
| Bread board and wires | Bud Industries | $7 |
| Arduino board | Egloo UNO R3 | $20 |
| LED (illumination source) | 3UTC-F | $2.68 |
| Spectral sensor (detector) | AMS AS7265-BLGT | $74.78 |
| | | Total cost: $104.46 |

device is not FDA-approved, and as a result, it is not meant for commercial or medical use, but rather for low-cost educational purposes. Table 2 lists the primary components that are required to assemble the device, as well as their individual and total costs. The total cost of this fluorometer is reduced as it eliminates the need for the dichroic mirror as well as excitation and emission filters. In addition, it is compact and lightweight and costs less compared to commercial high-end devices.

## 4. Conclusion

In summary, we proposed a smart fluorometer based on the principle of fluorescence spectroscopy. This device aims to detect lipid droplets accumulation in fluorescently labeled breast cancer cells. For this device to work properly, the light source and the optical sensor have been optimized. This fluorometer can detect changes in lipid droplet levels and therefore distinguish between moderately malignant and highly malignant cancer cell lines. In the current study, the results showed that the specificity and precision were 100% and 100%, respectively. We illustrate the use of our compact, lightweight, and portable fluorometer, which can be simply constructed from low-cost off-the-shelf components. This fluorometer potentially has a diverse range of possibilities. It can be utilized to perform fluorescence-based detection of several components in the cancer microenvironment and it will aid clinical and biomedical research.

## Supporting information

**S1 Fig. Emission signal resulted from BODIPY stained MCF7, MDA-MB-231.** (DOCX)

## Author Contributions

**Conceptualization:** Shiva Moghtaderi.

**Data curation:** Aditya Mandapati, Gerald Davies.

**Formal analysis:** Shiva Moghtaderi.

**Methodology:** Shiva Moghtaderi.

**Resources:** Aditya Mandapati.

**Supervision:** Khan A. Wahid, Kiven Erique Lukong.

**Validation:** Khan A. Wahid, Kiven Erique Lukong.

**Writing – original draft:** Shiva Moghtaderi, Aditya Mandapati.

**Writing – review & editing:** Shiva Moghtaderi, Khan A. Wahid, Kiven Erique Lukong.

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
