## [Decision Letter · Decision Letter 0]

17 Aug 2023

PONE-D-23-02819Smart and low-cost fluorometer for identifying breast cancer malignancy based on lipid droplets accumulationPLOS ONE

Dear Dr. Moghtaderi,

Thank you for submitting your manuscript to PLOS ONE. After careful consideration, we feel that it has merit but does not fully meet PLOS ONE’s publication criteria as it currently stands. Therefore, we invite you to submit a revised version of the manuscript that addresses the points raised during the review process.

We look forward to receiving your revised manuscript.

Kind regards,

Omnia Hamdy, PhD

Academic Editor

PLOS ONE

Journal Requirements:

"The New Frontiers in Research Fund (NFRF)"     

Reviewers' comments:

Reviewer's Responses to Questions

**Comments to the Author**

1. Is the manuscript technically sound, and do the data support the conclusions?

Reviewer #1: Partly

Reviewer #2: Yes

2. Has the statistical analysis been performed appropriately and rigorously? 

Reviewer #1: No

Reviewer #2: Yes

3. Have the authors made all data underlying the findings in their manuscript fully available?

Reviewer #1: No

Reviewer #2: Yes

4. Is the manuscript presented in an intelligible fashion and written in standard English?

Reviewer #1: Yes

Reviewer #2: Yes

5. Review Comments to the Author

Reviewer #1: Review_Manuscript Number: PONE-D-23-02819

The work presented through “Smart and low-cost fluorometer for identifying breast cancer malignancy based on lipid droplets accumulation” is highly interesting with overall quality photographs and compact presentation.

The claim “The goal of applying BODIPY probe is to target specific biological pathways at the cellular and molecular level. This means that the probes do not emit fluorescence unless they are fluorescently switched on. In other words, the dye is activated by specific cellular conditions or the presence of overexpressed receptors in cancer microenvironment.” is very important for the present work and should be justified in the text either through proper references or through further experiments.

Does BODIPY stain ordinary oil droplets also? If so, the efficacy of the fluorometer should be tested upon ordinary oil droplets as positive control.

Authors should elaborate the difference in the level of lipid droplets in normal breast cells and cancerous breast cells.

As a negative control, there should be a check on non-cancerous breast cells MCF10A to get a complete picture of the efficacy of the developed fluorometer.

Authors should also elaborate the proposed use of the developed device in real-life conditions, i.e., starting from the biopsy to the detection. What to detect? How to detect? How it is advantageous over the currently existing practices should be properly explained in the manuscript. For this, there should be a separate section along with a graphical presentation.

The data should be statistically verified and presented accordingly.

Reviewer #2: In this manuscript, the authors introduce a new fluorometer operating on fluorescence spectroscopy principles. The primary objective of their device is the identification of lipid droplet accumulation within fluorescently labeled breast cancer cells. The optimization of both the light source and optical sensor has been undertaken to ensure the proper functionality of this apparatus. By detecting fluctuations in lipid droplet levels, the fluorometer exhibits the capability to discern between moderately malignant and highly malignant cancer cell lines. Before arriving at a decision regarding publication, the reviewer wants to raise a few comments/issues:

1) While the proposed device offers substantial merit, its sensitivity appears comparatively low in relation to other existing devices. It would be valuable for the authors to elucidate scenarios in which this lower sensitivity remains acceptable or even advantageous.

2) The authors present the spectral response under various wavelengths for BODIPY-stained MCF7 and MDA-MB-231 cells. To bolster their argument, it would be prudent to also provide the spectral response under different wavelengths (instead of only one wavelength) for BODIPY unstained MCF7 and MDA-MB-231, enhancing the strength of their conclusions.

3) Fig. 5 and Fig. 6 appear to contain identical data and information. To avoid redundancy, it is recommended that the authors eliminate one of these figures.

4) To enhance visual clarity, the authors should consider employing distinct colors to differentiate between the two curves presented in Fig. 2.

6. PLOS authors have the option to publish the peer review history of their article (what does this mean?). If published, this will include your full peer review and any attached files.

Reviewer #1: No

Reviewer #2: No

---

## [Author Response · Author response to Decision Letter 0]

11 Oct 2023

Reviewer #1: 

The work presented through “Smart and low-cost fluorometer for identifying breast cancer malignancy based on lipid droplets accumulation” is highly interesting with overall quality photographs and compact presentation.

Reviewer Comment: The claim “The goal of applying BODIPY probe is to target specific biological pathways at the cellular and molecular level. This means that the probes do not emit fluorescence unless they are fluorescently switched on. In other words, the dye is activated by specific cellular conditions or the presence of overexpressed receptors in cancer microenvironment.” is very important for the present work and should be justified in the text either through proper references or through further experiments.

Answer: We agree with this feedback. The paragraph has been reworded to reflect the use of BODIPY as a fluorescent dye that was used due to its binding to neutral lipid droplets in cells. Changes made to page 4, line 4 onwards. 

Reviewer comment: Does BODIPY stain ordinary oil droplets also? If so, the efficacy of the fluorometer should be tested upon ordinary oil droplets as positive control.

Answer: Yes, BODIPY can stain oil droplets and has been used as such previously. However, as we are using a cell-based system, the proposed use of ordinary oil droplets as a positive control may not be appropriate. 

Reviewer comment: Authors should elaborate the difference in the level of lipid droplets in normal breast cells and cancerous breast cells.

Answer: The differences between MCF10A cells (normal breast epithelium cells) vs breast cancer cells (MCF7 and MDA-MB-231) have been elaborated on more specifically on page 2, line 25 onwards.

Reviewer comment: As a negative control, there should be a check on non-cancerous breast cells MCF10A to get a complete picture of the efficacy of the developed fluorometer.

Answer: We agree with this feedback and have therefore included BODIPY-stained MCF10A samples as part of the revised manuscript.

Reviewer comment: Authors should also elaborate the proposed use of the developed device in real-life conditions, i.e., starting from the biopsy to the detection. What to detect? How to detect? How it is advantageous over the currently existing practices should be properly explained in the manuscript. For this, there should be a separate section along with a graphical presentation.

Answer: The potential applications of our proposed fluorometer extend prominently to the domains of early detection and screening. This include complementing the advancements in wireless capsule endoscopy that introduces a new dimension of assessing tissue characteristics by considering the lipid droplets accumulation.

In the context of early detection and screening, our fluorometer's ability to detect cancer malignancy through lipid droplet accumulations holds promise for providing a non-invasive and cost-effective method for identifying cancerous cells at an early stage. This can play a crucial role in identifying potential cases for further accurate evaluations.

Reviewer comment: The data should be statistically verified and presented accordingly.

Answer: The statistical evaluation including the calculation of sensitivity, accuracy, specificity and precision have been elaborated on more specifically on page 7. Fig. 9 also demonstrates the confusion matrix drawn from the measurements obtained with our prototype.

Reviewer #2:

In this manuscript, the authors introduce a new fluorometer operating on fluorescence spectroscopy principles. The primary objective of their device is the identification of lipid droplet accumulation within fluorescently labeled breast cancer cells. The optimization of both the light source and optical sensor has been undertaken to ensure the proper functionality of this apparatus. By detecting fluctuations in lipid droplet levels, the fluorometer exhibits the capability to discern between moderately malignant and highly malignant cancer cell lines. Before arriving at a decision regarding publication, the reviewer wants to raise a few comments/issues:

Reviewer Comment: While the proposed device offers substantial merit, its sensitivity appears comparatively low in relation to other existing devices. It would be valuable for the authors to elucidate scenarios in which this lower sensitivity remains acceptable or even advantageous.

Answer: The reviewer's observation regarding the sensitivity of our proposed device compared to existing technologies is acknowledged. It's important to understand that, despite the fact that our device's sensitivity may seem comparatively lower, it should be assessed in light of both the intended use and the larger clinical landscape. Our device's affordability and portability make it suitable for deployment in regions with limited resources. In such settings, even a device with slightly lower sensitivity can play a crucial role in identifying potential cases for further accurate evaluations, thus contributing to early intervention. Moreover, in cases where frequent monitoring is required, such as post-treatment follow-up, a less sensitive device may still provide valuable insights. It can help in tracking changes over time and alerting healthcare providers to any deviations from the baseline.

Reviewer Comment: The authors present the spectral response under various wavelengths for BODIPY-stained MCF7 and MDA-MB-231 cells. To bolster their argument, it would be prudent to also provide the spectral response under different wavelengths (instead of only one wavelength) for BODIPY unstained MCF7 and MDA-MB-231, enhancing the strength of their conclusions.

Answer: We have studied the emitted fluorescence signal under four different LED wavelengths to compare the resulting signals from MDA-MB-231 slides, MCF7 slides and MCF10A in section 3. The average fluorescence intensity of 52 slides captured by the spectral detector under varying excitation wavelengths is observed in Fig. 5.

Reviewer Comment: Fig. 5 and Fig. 6 appear to contain identical data and information. To avoid redundancy, it is recommended that the authors eliminate one of these figures.

Answer: We agree with this feedback. We have eliminated figure 6.

Reviewer Comment: To enhance visual clarity, the authors should consider employing distinct colors to differentiate between the two curves presented in Fig. 2.

Answer: We agree with this feedback and have carefully considered your suggestions and made the necessary revisions to Fig. 2.

---

## [Decision Letter · Decision Letter 1]

14 Nov 2023

Smart and low-cost fluorometer for identifying breast cancer malignancy based on lipid droplets accumulation

PONE-D-23-02819R1

Dear Dr. Moghtaderi,

We’re pleased to inform you that your manuscript has been judged scientifically suitable for publication and will be formally accepted for publication once it meets all outstanding technical requirements.

Kind regards,

Omnia Hamdy, PhD

Academic Editor

PLOS ONE

Additional Editor Comments (optional):

Reviewers' comments:

Reviewer's Responses to Questions

**Comments to the Author**

1. If the authors have adequately addressed your comments raised in a previous round of review and you feel that this manuscript is now acceptable for publication, you may indicate that here to bypass the “Comments to the Author” section, enter your conflict of interest statement in the “Confidential to Editor” section, and submit your "Accept" recommendation.

Reviewer #1: All comments have been addressed

Reviewer #2: All comments have been addressed

2. Is the manuscript technically sound, and do the data support the conclusions?

Reviewer #1: Yes

Reviewer #2: Yes

3. Has the statistical analysis been performed appropriately and rigorously? 

Reviewer #1: Yes

Reviewer #2: Yes

4. Have the authors made all data underlying the findings in their manuscript fully available?

Reviewer #1: Yes

Reviewer #2: Yes

5. Is the manuscript presented in an intelligible fashion and written in standard English?

Reviewer #1: Yes

Reviewer #2: Yes

6. Review Comments to the Author

Reviewer #1: The authors have adequately addressed the issues raised. The manuscript in its current version can be accepted for publication.

Reviewer #2: (No Response)

7. PLOS authors have the option to publish the peer review history of their article (what does this mean?). If published, this will include your full peer review and any attached files.

Reviewer #1: No

Reviewer #2: No

---

## [Editor Report · Acceptance letter]

11 Dec 2023

PONE-D-23-02819R1 

Smart and low-cost fluorometer for identifying breast cancer malignancy based on lipid droplets accumulation 

Dear Dr. Moghtaderi:

I'm pleased to inform you that your manuscript has been deemed suitable for publication in PLOS ONE. Congratulations! Your manuscript is now with our production department. 

Kind regards, 

on behalf of

Dr. Omnia Hamdy 

Academic Editor

PLOS ONE